# Optimization of Methods for the Production and Refolding of Biologically Active Disulfide Bond-Rich Antibody Fragments in Microbial Hosts

**DOI:** 10.3390/antib9030039

**Published:** 2020-08-05

**Authors:** Bhupal Ban, Maya Sharma, Jagathpala Shetty

**Affiliations:** 1Antibody Engineering and Technology Core, University of Virginia, Charlottesville, VA 22904, USA; 2Department of Cell Biology, University of Virginia, Charlottesville, VA 22904, USA; 3Pharmaceutical Biotechnology Center, Indiana Biosciences Research Institutes (IBRI), Indianapolis, IN 46202, USA; 4Department of Data Science, School of Informatics and Computing Indiana University–Purdue University Indianapolis (IUPUI), Indianapolis, IN 46202, USA; mayashar@iu.edu; 5Department of Biomedical Engineering, University of Virginia, Charlottesville, VA 22904, USA

**Keywords:** recombinant antibodies, inclusion bodies, refolding, redox pair, disulfide bonds

## Abstract

Antibodies have been used for basic research, clinical diagnostics, and therapeutic applications. *Escherichia coli* is one of the organisms of choice for the production of recombinant antibodies. Variable antibody genes have canonical and non-canonical disulfide bonds that are formed by the oxidation of a pair of cysteines. However, the high-level expression of an antibody is an inherent problem to the process of disulfide bond formation, ultimately leading to mispairing of cysteines which can cause misfolding and aggregation as inclusion bodies (IBs). This study demonstrated that fragment antibodies are either secreted to the periplasm as soluble proteins or expressed in the cytoplasm as insoluble inclusion bodies when expressed using engineered bacterial host strains with optimal culture conditions. It was observed that moderate-solubilization and an in vitro matrix that associated refolding strategies with redox pairing more correctly folded, structured, and yielded functionally active antibody fragments than the one achieved by a direct dilution method in the absence of a redox pair. However, natural antibodies have canonical and non-canonical disulfide bonds that need a more elaborate refolding process in the presence of optimal concentrations of chaotropic denaturants and redox agents to obtain correctly folded disulfide bonds and high yield antibodies that retain biological activity.

## 1. Introduction

Currently, monoclonal antibodies (mAbs) are key biologics for several applications such as basic science research, therapeutics protein, and clinical diagnostics assays such as immunoassays, immunohistochemistry, flow cytometry, and immuno-PET scanning. In addition, antibodies are used in proteome research for the identification of biomarkers [1]. Advancements in the area of immunoinformatic and recombinant antibodies have changed the applications of domain antibodies such as the single chain fragment variable (scFv), the antigen binding fragment (Fab), and single domain of variable heavy homodimers (VHH) are very popular reagents for basic scientific and clinical applications [2]. Evidently, the nature of the antibody sequence and structural variability of Abs is necessity to obtain efficient reagents that are possibly also simple to use. At the same time, it implies the necessity of developing customized expression and purification procedures to obtain enough yield. A variety of expression hosts are used for the production of recombinant antibodies using bacterial, mammalian cell, yeast, and insect cells.

Among these selection hosts, the bacterial expression host system, generally, is the desirable host due to its well-founded genetics, properties of fast growth rate with high density, inexpensive substrates, simple cultivation, and availability of large numbers of cloning vectors along with engineered bacterial host strains with high production yields [3]. Though rich disulfide bond protein requires a sophisticated folding apparatus under oxidizing environments for the generation of disulfide bonds in bacterial systems, particularly in the periplasmic space, it allows the correct formation of disulfide bonds and assembly to a functional variable fragment with a maximum antigen-binding capacity, as IgG [4]. Due to the contrasting molecular size, production cost, and time consumption, fragments such as the single-chain fragment (scFv) [5], camelid, llama (VHH) [6], and shark IgNAR [7] are therefore amenable to production at a low cost in microbial systems. However, the periplasmic expression of recombinant proteins has two major drawbacks of low yield and frequent aggregation. Alternatively, the production of domain antibodies using fusion constructs in tandem with a stretch of amino acid (peptide tag) or a large polypeptide (Thioredoxin (Trx), glutathione S-transferase (GST), and maltose-binding protein (MBP) has been demonstrated in bacterial periplasmic and cytoplasmic systems [8]. Unfortunately, the reducing environment in the cytoplasmic region remains a limiting factor for the folding of the majority of cysteine rich proteins such as antibodies. An approach to overcome the shortcoming is to express the antibodies in the cytoplasm of mutant *E. coli* strains that provide an oxidizing environment. However, often, the formation of a partially folded/misfolded protein is triggered, which is forced to aggregate as non-functional inclusion bodies (IBs) [9]. Largely, cysteine-rich proteins are formed as IBs in both the cytoplasmic and periplasmic compartments of *E. coli* [10]. Utilizing various strategies for molecular and protein engineering such as gene fusion technology, co-production of chaperones [11], use of engineered host *E.coli* strains [12], optimizing the growth conditions such as temperature, inducer concentration, and induction time, can often help in increasing the formation of efficient soluble protein production [13].

Recently, it has been reported that recombinant proteins were overexpressed in the soluble form in the bacterial expression system by adding compatible solutes (chemical additives) such as sorbitol, arginine, and trehalose in the expression medium [14]. Although proteins with disulfide bonds are also found in the cytoplasmic region of both eukaryotes and prokaryotes, most of them are transient and involve cysteines located in redox-active sites of enzymes [15]. Both classical and non-classical IBs contain relatively pure and intact recombinant proteins, and several approaches have been reported to recover biologically active proteins from these aggregated forms. Usually, during an in vitro refolding procedure, IBs are denatured by dissolving with a high or mild concentration of chaotropic denaturants such as urea, guanidine hydrochloride (GdnHCl) [16], or alternately with non-ionic or ionic detergents, such as Triton X-100, N-lauroylsarcosine, dimethylsulfoxide (DMSO) and a low percentage (~5%) of n-propanol [17]. In addition, dithiothreitol (DTT) or 2-mercaptoethanol was added to reduce undesirable inter- and/or intra-molecular disulfide bonds [18]. Different robust refolding techniques have been reported which work in conjunction with direct dilution, diafiltration, chromatography including size exclusion chromatography, matrix- or affinity-based techniques, hydrophobic interaction chromatography [19,20], or non-chromatographic strategies [21]. In particular, the impact of protein stabilizing or destabilizing low- and high-molecular-weight additives, as well as micellar and liposomal systems on protein refolding is utilized in these methods. Likewise, techniques mimicking the principles encountered during in vivo folding such as processes based on natural and artificial chaperones [22] and optimal concentrations of both oxidizing and reducing agents [23] and propeptide-associated [24] protein refolding, have also been applied.

In this study, we have demonstrated the production of rich cysteine residue antibodies using the microbial host with a high yield and function. Antibodies were initially expressed in a partially soluble form. The disulfide bond rich antibodies: Rab-scFv, Hum-scFv, and VHH were firstly expressed in both periplasmic and cytoplasmic space and successfully purified using affinity chromatography. The misfolded antibodies from the inclusion bodies were properly refolded by a matrix-associated refolding method.

Firstly, the inclusion bodies were solubilized in a buffer containing a denaturing agent and, the solubilized unfolded His-tag antibodies were bound to an immobilized metal affinity matrix (IMAC) nickel affinity matrix support. The bound unfolded protein was then allowed to refold by gradually replacing the denaturing buffer with a native buffer. It was observed that moderate-solubilization and in vitro matrix-associated refolding strategies with redox pairing yielded functionally active antibodies that were superior to those obtained by in vitro refolding methods such as the direct dilution method.

## 2. Materials and Methods

### 2.1. Bacterial Strains

*E. coli BL*21, *B F’ ompT gal dcm lon hsdSB(rB–mB–) [malB+]K-12(λS)*, is a widely used non-T7 expression cells. *E. coli* BL21(DE3), *B F’ ompT gal dcm lon hsdSB(rB–mB–) λ(DE3 [lacI lacUV5-T7p07 ind1 sam7 nin5]) [malB+]K-12(λS)* and Shuffle^®^ T7 Express; T7 expression/K12 strain, Genotype: *fhuA2 lacZ: T7 gene1 [lon] ompT ahpC gal λatt: pNEB3-r1-cDsbC (SpecR, lacIq) ΔtrxB sulA11 R(mcr-73::miniTn10--TetS)2 [dcm] R(zgb-210::Tn10 --TetS) endA1Δgor ∆(mcrC-mrr)114::IS10* (NEB’s strain), Rosetta-gami B, *(F’ ompT hsdSB (rB– mB–) gal dcm lacY1 ahpC (DE3) gor522::Tn10 trxB pRARE (Cam^R^, Kan^R^, Tet^R^)* (New England BioLabs, US) were purchased for this experiment. All *E. coli* strains were propagated in Luria-Bertani (LB) broth (tryptone 10.0 g/L, yeast extract 5.0 g/L, Sodium Chloride (NaCl) 5.0 g/L, pH 7.2–7.5), supplemented with ampicillin (100 μg/mL) or tetracycline (100 μg/mL).

### 2.2. Chemicals and Reagents

Sodium chloride, glycerol, sodium phosphate saline (PBS), isopropyl β-d-1-thiogalactopyranoside (IPTG), Tris HCL and base, Glycine, lysozyme, dithiothreitol (DTT), and ethylene-di-aminetetraacetic acid (EDTA), Coomassie G-250 were purchased from VWR life science. Inclusion body (IB) solubilizing and protein refolding reagents: GSH, GSSG, L-Arginine, and urea were from Sigma-Aldrich (St. Louis, MI, USA)). PCR cloning kit: PET series vectors; all restriction enzymes; T4 DNA ligase and rapid protein assay BCA kit were from New England Biolabs (NEB, Ipswich, MA, USA) and Thermo Scientific (Cincinnati, OH, USA). ELISA reagents: Human serum albumin (HAS), bovine serum albumin (BSA), Tween-20, Citric acid, H_2_O_2_, o-phenylenediamine dihydrochloride (OPD) were from Sigma-Aldrich and Alfa Aesar. Protein purification apparatus and affinity chromatography columns were purchased from GE Healthcare.

### 2.3. Isolation, Design, and Synthesis of Antibody Genes

The human serum albumin (HSA) specific single domain antibody (VHH) was isolated from the immune llama derived phage display library. The immune library was gifted from the University of California (UC devise), and surface display bio-panning was conducted at the University of Virginia Core center against albumins from different species [25]. For this study, we selected high cysteine conserved VHH antibody sequences. The rabbit hybridoma-derived monoclonal antibody gene was isolated from hybridoma cells. Briefly, the rabbit was immunized with a peptide-KLH conjugated antigen, and a hybridoma fusion was carried out by utilizing immune spleen cells with rabbit-based myeloma cells [26]. Several hybridoma clones were screened, and clones with a desirable epitope against the target peptide were selected. The isolation of rabbit variable heavy and light chain genes was achieved using specific primers, and they were converted to an scFv format [27]. The human-derived anti-Ebola virus (Kz52) heavy and light chain variable genes were obtained from an Ebola virus glycoprotein in a complex with a neutralizing antibody crystal structure ID: 3CSY at Protein Data Bank (PDB) [28]. The human and rabbit antibody sequences of heavy and light chains of variable regions were codon-optimized, and the corresponding DNAs were synthesized in an scFvs format using a 20 aa (GGGGS) 4 linker in between variable heavy and light chains. Likewise, the llama derived antibody VHH gene was synthesized with appropriate cloning restriction sites at 5 and 3′, respectively. The antibody variable genes were denoted as Rab-scFv, Hum-scFv, and VHH, respectively. The structural integrity of the newly designed Rab-scFv, Hum-scFv, and VHH antibody molecules was verified by in silico homology modeling. The canonical and non-canonical disulfide bonds in variable regions were predicted.

### 2.4. Designing and Cloning of Abs Genes with and without a Fusion Tag

#### 2.4.1. Construction of Plasmids for Periplasmic Expression of Antibodies

The rabbit, human, and llama derived antibody genes were synthesized, and then sub-cloned to a pET-based vector (PAB- cMyc-His tag) using *BssHII* and *NheI* restriction enzymes in such a way to append a C-terminal cMyc tag for detection and a 6xHis-tag for purification by metal affinity chromatography (Figure 1a). The vector has a pelB leader sequence with a T7 promotor. The equivalent plasmids clones were designated as the rabbit gene: RabscFv-pAB, human gene: HumscFv-pAB, and llama gene: VHH-pAB. The DNA sequences of the plasmids were confirmed by sequencing.

#### 2.4.2. Construction of Plasmids for Cytosolic Expression of Antibodies

The synthetic rabbit scFv, human scFv, and VHH genes were PCRs amplified with specific primers that were flanked by BamHI and XhoI restriction sites. The VHH gene was then sub-cloned into pET32(b) including a thioredoxin tag (TrxA) at N-terminus (designated as VHH-pET32(b)), and pET-24(a) (designated as VHH-pET24(a)) for cytosolic bacterial expression, as illustrated in Figure 1b,c, respectively. The rabbit and human antibody genes were cloned into a pET-24(a) vector, and resulting clones were designated as RabscFv-pET24(a) and HumscFv-pET24(a) as illustrated in Figure 1c. These vectors contained T7 promotor for controlling protein expression (Novagen, Madison, WI, USA) along with a cMyc tag for detection, and a 6xHis-tag at the C-termini for purification, by metal affinity chromatography. The recombinant constructs were further verified by sanger sequencing.

### 2.5. Expression and Extraction of scFvs, VHH Recombinant Antibodies from Periplasmic and Cytoplasmic Regions

The E. coli strains, such as BL21(DE3), Shuffle T7 Express, and Rosetta-gami B, were transformed with nine plasmids, namely RabscFv-pAB, HumscFv-pAB, VHH-pAB, RabscFv-pET32(b), HumscFv-pET32(b), VHH-pET32(b), RabscFv-pET24(a), HumscFv-pET24(a), and VHH-pET24(a). The transformed bacterial clones were grown overnight in LB agar plates in the presence of ampicillin for both pAB and pET32(b), and in the presence of kanamycin for pET24(a) at 37 °C. Single clones from each plate were then inoculated in different culture media such as LB, 2xYT, and TB, supplemented with ampicillin (100 µg/mL), and/or with 50 µg/mL kanamycin. The primary cultures grown overnight were re-inoculated in fresh media with corresponding antibiotics and allowed to grow at 37 °C until the OD_600_, reached 0.5. All the secondary cultures grown at 37 °C were then split into three groups and were induced with 0.3 mM IPTG and grown at 18, 28, and 30 °C overnight. The cell pellets were then harvested by centrifugation at 4000× *g*, and cell pellets were resuspended with lysis buffer containing 10 mM HEPES, pH 7.4, 1 mg/mL hen egg lysozyme, and an EDTA-free protease inhibitor cocktail (Halt protease inhibitor cocktail, Thermo Fisher Scientific). The cells were incubated on ice for 1 h with shaking rotor. After incubation, MgCl_2_ and DNase-I were added to a final concentration of 10 mM and 20 U/mL, respectively. The cells were incubated at room temperature (RT) for 30 min and pelleted by centrifugation for 20 min at 4 °C at 18,000× *g*. The crude supernatants were collected, and the pellets were resuspended with an equal volume of lysis buffer (20 mM Hepes buffer containing 8 M urea or 6 M guanidine hydrochloride). The expression levels of antibodies in supernatants as soluble Abs and pellets as insoluble Abs were analyzed using Western blot analysis after separating on SDS-PAGE.

### 2.6. Western Blot Analysis

The cell pellets were utilized for the detection of expressed antibodies in periplasmic and cytosolic compartments, and further for their presence in soluble and insoluble fractions. The resultant soluble and insoluble proteins in SDS-PAGE loading buffer containing 100 mM DTT were heated to 95 °C for 10 min and loaded onto 4–20% gradient SDS-PAGE. After SDS-PAGE, the proteins were transferred to a 0.45 µm pore size nitrocellulose membrane (Amersham™ Protran™, GE Healthcare) using a Trans Blot^®^ SD Semi-Dry Transfer Cell (BIO-RAD) at 0.5 A for 1 h. The membranes were incubated in PBS with 5% skim milk powder and 0.1% Tween 20 for 2 h. After washing with PBS with 0.05% Tween 20 (PBST), the antibody such as anti-c myc- horseradish peroxidase (HRP) conjugate was added at a dilution of 1:3000 in PBS containing 0.1% Tween 20 for 1 h at room temperature (RT) under constant slow rocking. After washing with PBST, specific immunoreactive bands were visualized using Tetramethylbenzidine (TMB) substrate.

### 2.7. Dot-Blot Analysis

The bacterially expressed and purified antibody fragments were analyzed in native conditions using dot-blot analysis. The proteins were spotted on a 0.45 µm pore size nitrocellulose membrane (Amersham™ Protran™, GE Healthcare Chicago, IL, USA) and allowed to dry at room temperature for 1 h. The membranes were incubated in PBS with 5% skim milk powder and 0.1% Tween 20 for 2 h. After washing with PBS with 0.05% Tween 20 (PBST), the antibody such as anti-c Myc-HRP conjugate was added at a dilution of 1:3000 in PBS containing 0.1% Tween 20 for 1 h at room temperature (RT) under constant slow rocking. After washing with PBST, specific immunoreactive bands were visualized using TMB substrate (Tetramethylbenzidine substrate, KPL, Indianapolis, IN, USA).

### 2.8. Expression and Preparation of Soluble Abs from Periplasmic and Cytoplasmic Compartments

The frozen pellets resulting from 2 L of culture were briefly thawed and suspended in 200 mL of lysis buffer containing 1 mg/mL lysozyme in PBS plus 0.1% Triton 100 and EDTA free protease inhibitor cocktail (ThermoScientific, Waltham, MA, USA). The lysis mixture was incubated on ice for an hour, and then 10 mM MgCl_2_ and 1 μg/mL DNaseI were added and the mixture was incubated at RT for 30 min. The final lysate mixture was centrifuged at 12,000× *g* for 30 min and the supernatants were collected. The periplasmic extract was analyzed using SDS-PAGE and Western blot analysis using an anti-His/or c-Myc tag antibody before loading onto the chromatography system. Soluble antibody fragments, both rabbit and human scFvs and VHH in the supernatant, were either purified via Ni-NTA (cOmplete™ His-tag Purification Resin, Roche, St louis, MI, USA) or protein L (GE life science, Marlborough, MA, USA) affinity chromatography. Briefly, the cleared lysate containing the antibody was diluted with equilibrium buffer-like lysis buffer (50 mM NaH_2_PO_4_, 300 mM NaCl, 10 mM imidazole, pH 8.0) and applied to gravity-flow columns (2 mL bed volume) equilibrated with lysis buffer. The columns were washed twice with Ni-NTA wash buffer (50 mM NaH_2_PO_4_, 300 mM NaCl, 20 mM imidazole, pH 8.0). Proteins bound to the Ni-NTA material were eluted with elution buffer (50 mM NaH_2_PO_4_, 300 mM NaCl, 250 mM imidazole, pH 8.0). On the other hand, protein L-bound scFvs were eluted with 0.1 M glycine, pH 3.0. The pH of the solution was neutralized with 1 M Tris-HCl, pH 8.0. The final eluted samples were buffer exchanged with the appropriate buffer such as PBS for final assay parameters.

### 2.9. Preparation of Inclusion Bodies for Isolation of Refolding Antibody Fragments

The primary culture of all scFvs recombinant clones, RabscFv-pET24(a), HumscFv-pET24(a), and VHH-pET24(a) was used for high expression in 2xYT and terrific broth (TB), as described above, at 25 °C overnight under 0.3 mM IPTG induction. The cells were harvested with centrifugation at 4000× *g* for 20 min. The pellet was resuspended in PBS buffer (20 mM PBS, 100 mM NaCl, pH 7.4) containing 200 µg/mL lysozyme and protease inhibitors. The cells were incubated for about 1 h in the cold room in an end to end shaker. When the lysate appeared sticky due to the release of DNA, about 20 μg/mL of DNase I was added along with 10 mM MgCl_2_, mixed thoroughly, and left in RT with shaking for about 20 min to make sure that all the DNA was digested. Cell lysates were centrifuged at 15,000× *g*, at 4 °C, for 30 min. The supernatant containing the soluble protein was separated. The pellet was used for the isolation of inclusion bodies. The insoluble pellet was resuspended in inclusion body (IB) wash buffer (20 mM PBS, 500 mM NaCl, 1 mM ethylenediaminetetraacetic acid, 2 M urea, 2% *v*/*v* Triton X–100, pH 7.4). The suspension was washed vigorously for 30 min at room temperature. The pellet was washed one more time with the above buffer without urea. A final wash with PBS buffer was implemented. The resulting pellet (inclusion bodies) can be saved at −20 °C until further use.

#### 2.9.1. Denaturation and Solubilization of Antibody from Inclusion Bodies

The canonical and non-canonical inclusion bodies (IBs) were thawed and resuspended in solubilization buffer (20 mM PBS, 500 mM NaCl, 6 M urea, 1 mM dithiothreitol, 10 mM imidazole, pH 8.2) and stirred vigorously for 3 h at RT. The dissolved proteins were then centrifuged for 30 min at 15,000× *g*, 4 °C, and the supernatant containing solubilized IBs were collected in a tube. The collected supernatants were checked on SDS-PAGE electrophoresis and dot-blot analysis. The solubilized inclusion bodies (IBs) containing antibodies verified in the supernatants were further used for refolding experiments.

#### 2.9.2. Refolding by Dialysis, and Purification of Refolded Solubilized Antibodies from Inclusion Bodies (IBs)

The resuspended antibody-protein material was then diluted 50% in dialysis buffer #1 (20 mM PBS, 500 mM NaCl, 2 M urea, 1 mM dithiothreitol, 10 mM imidazole, pH 8.0) resulting in a 4 M urea containing solution. The protein solution was then dialyzed overnight at 4 °C in snakeskin dialysis tubing (Pierce) against 2 L of buffer #1. The following day, the dialysis buffer was changed to 2 L of dialysis buffer #2 (20 mM PBS, 500 mM NaCl, 1 M urea, 1 mM dithiothreitol, 10 mM imidazole, pH 8.0) for overnight dialysis at 4 °C. The following day, the dialysis buffer was diluted 50% with water and dialysis continued overnight. Any insoluble material was centrifuged using 18,000× *g* at 2–8 °C for 20 min, and the remaining protein solution was dialyzed overnight at 4 °C against 1 L of dialysis buffer #3 (20 mM PBS, 500 mM NaCl, 1 mM dithiothreitol, 10 mM imidazole, pH 8.0) to remove the remaining urea. The final dialyzed protein solution was clarified by centrifugation using 18,000× *g* at 2–8 °C for 20 min and the supernatant was loaded onto a cOmplete His-tag purification resin (Roche) column pre-equilibrated with equilibration buffer (20 mM PBS, 500 mM NaCl, 1 mM dithiothreitol, 10 mM imidazole, pH 8.0). The column was washed with an excess of wash buffer (20 mM PBS, 500 mM NaCl, 10 mM imidazole, pH 8.0, 0.1% Triton X-100) followed by washing with equilibration buffer to remove the Triton X-100. The refolded antibody was eluted in 1 mL aliquots from the resin using elution buffer (20 mM PBS, pH 8, 250 mM Imidazole). All the protein samples were resolved on 4%–20% gradient SDS-PAGE gel and visualized by staining with Coomassie-blue. Specific scFvs and VHH bands were analyzed by comparing them with the prestained protein marker (Bio-Rad).

#### 2.9.3. Refolding and Purification of Solubilized Antibodies from IBs Using the Matrix-Associated Method

The inclusion bodies were resuspended in solubilization buffer (20 mM PBS, 500 mM NaCl, 1 mM DTT, 6 M urea, 10 mM imidazole, including 400 mM L-arginine (Sigma, A5006), 3 mM reduced glutathione (GSH), 0.9 mM oxidized glutathione (GSSG)) pH 8.0). The solubilized IBs were cleared by centrifuging at 15,000× *g* for 30 min. The supernatant containing the antibody was subjected to a matrix-associated or affinity column-based protein refolding method that was achieved through the elimination of the chaotropic agent using a linear gradient with refolding buffers. The antibody samples were loaded onto a column equilibrated in equilibration buffer (same as the solubilization buffer above). The affinity bound antibody was allowed to fold properly by gradually washing with 5 column volume (cv) each of 2 liner gradient wash buffers: Wash buffer 1 (20 mM PBS, 500 mM NaCl, 1 mM DTT, 10 mM imidazole, 400 mM L-arginine, 3 mM reduced glutathione (GSH), 0.9 mM oxidized glutathione (GSSG)) pH 8.0) with decreasing amounts of urea (gradient of 6 to 0 M) followed by wash buffer 2 (20 mM PBS, 500 mM NaCl, 10 mM imidazole) pH 8.0 with decreasing amounts of DTT, GSH, GSSG, and L-arginine. The column was washed with an excess of wash buffer (20 mM PBS, 500 mM NaCl, 10 mM imidazole, pH 8.0) The refolded antibody was eluted in 1 mL aliquots from the resin using elution buffer (20 mM PBS, 250 mM Imidazole, pH 8). The antibody quality was analyzed by SDS-PAGE as described above. Fractions containing eluted protein were pooled and concentrated by using Millipore-Amicon 15 mL ultra-centrifugal filter units (10 kDa MWCO).

### 2.10. ELISA Analysis of Antibody Binding to Target Antigens

Binding analysis of solubilized and refolded scFv and VHH antibodies were performed by direct ELISA using 96 well fat bottom polystyrene plates. The commercially available recombinant version of the capsid glycoprotein (GP) of Ebola GP [29] and human serum albumin (HSA) were coated with equimolar amounts at 4 °C overnight. The unbound protein samples were washed with PBST buffer (PBS with 0.05% Tween 20), and the untenanted spaces were blocked with 3% (*w*/*v*) non-fat skim milk dissolved in PBS. Several dilutions of scFvs and VHH were used as primary antibody and plates were incubated for 1 h at room temperature. Plates were washed three times and incubated with anti-cMyc–horseradish peroxidase conjugate as a secondary antibody for 1 h at RT. HRP activities were measured at 450 nanometer using TMB substrate (tetramethylbenzidine substrate, KPL) and readings were taken using an ELISA plates reader (EL-808) Biotek.

## 3. Results

### 3.1. Bioinformatics Analysis of Canonical and Non-Canonical Disulfide Bonds

The genes of the variable heavy chain and variable light chain regions from human-derived anti-Ebola (HumscFv), rabbit derived anti-peptide specific antibody (RabscFv), and llama derived anti-human serum albumin (HSA) were obtained from different sources as described in method Section 2.3. Hence, given antibody sequences were annotated and, canonical and non-canonical disulfide bond formation in the variable heavy chain (VH) and variable light chain (VL), were predicted including free cysteine residues. Based on this approach, we generated models for the variable domains of scFvs (rabbit and human) and VHH genes using the antibody prediction tool SAbPred model (http://opig.stats.ox.ac.uk) (Figure 2). We analyzed classical cysteine residues in VH and VL genes including high cysteine insertion and cysteine state and disulfide bond partner prediction using the DiANNA 1.1 web server as illustrated (Figure 2). These antibody sequences predicted cysteines at positions C^22^–C^92^ for VH and C^23^–C^88^ for VL of human genes (1c), C^21^–C^94^ for VH, C^23^–C^88^ for VL of rabbit genes (2b), and C^22^–C^96^ for the camelid single-domain (VHH) antibody (Figure 2a) as canonical bonds. Additionally, the cysteine residues at positions C^34^–C^59^ for VH, at C^80^–C^90^ for VL of rabbit gene, and C^50^-C^109^ for camelid VHH gene were predicted to generate non-canonical disulfide bonds as shown in Figure 2a,b.

### 3.2. Expression and Optimization of Recombinant Antibody Production

The single chain antibody fragments such as HumscFv, RabscFv, and single domain VHH proteins with/without fusion thiol-disulfide exchange protein TrxA were efficiently expressed in the microbial system using the T7 lac promotor containing a vector with/without a bacterial pelB signal peptide using different engineered bacterial strains in periplasmic and cytoplasmic compartments in different growth conditions (Figure 3a,b).

Under normal expression conditions, none of the verified human and rabbit scFvs and VHH with, or without, the TrxA tag were expressed in a soluble form. The rabbit scFv expression in different *E. coli* strains was also analyzed by lowering expression temperature and inducer IPTG concentration. It was observed that induced cultures resulting in lower biomass (cell density) yielded generally more soluble antibodies (Figure 4a,b). A slight relationship between the induction and fermentation temperature was noticed, i.e., early induction coupled with low fermentation temperatures resulted in as much as 60% antibody in a soluble form (Figure 4a). Lowering the temperature was also associated with lower biomass density in different bacterial strains (Figure 4b–d) due to slow bacterial metabolism. Overall, an optimal temperature for induction and culture to produce soluble RabscFv and HumscFv protein production was observed to be at around 20–25 °C (Figure 3). Notably, we observed scFv expressed in *E. coli* SHuffle^®^ T7 express and B1 21(DE)3 bacterial strains at different temperatures and observed white-colored pellets at 18 °C, which became a slightly brownish and blackish color at 25 and 30 °C, (Figure 4c,d), respectively.

For cytosolic expression of scFvs and VHH antibodies, the genes were constructed with/without the thiol-disulfide exchange protein TrxA at N-terminus, and efficiently expressed in the microbial system under a T7 lac promotor-vector without bacterial pelB leader peptide. The overall soluble antibody productivity in both periplasmic and cytoplasmic spaces are presented in Figure 5. We observed a significantly higher expression of soluble proteins (with or without non-canonical disulfide bonds) in engineered SHuffle^®^ T7 in both periplasmic and cytoplasmic regions as monitored by Western blot analysis (Figure 5a–d,g,h). However, due to the reducing condition of the *E. coli* cytoplasmic compartment, expressed scFvs and VHH are hardly capable of forming disulfide bonds and produced insoluble aggregates as IBs [30]. Unfortunately, the human, rabbit scFvs, and VHH expression in soluble fractions did not show any improvements with altered expression conditions.

It has been reported earlier that scFv and VHH antibody fragments are efficiently made in the soluble form when expressed as fusion proteins with TrxA/B. Accordingly, here, a significant level of antibody expression in a soluble fraction was observed (Figure 5c,d). However, we also noticed that the purified VHH-TrxA protein significantly lost biological activity, such as the binding activity in an ELISA assay (Figure 5e). In addition, we evaluated soluble antibody production using two scFv constructs: one containing canonical disulfide bonds (HumscFv) and the other containing non-canonical disulfide bonds (RabscFv), (Figure 5f,g). We noticed that the relative increase in the number of non-canonical disulfide bonds had a concomitant negative effect on the production of soluble proteins in both compartments, mostly in the cytoplasmic region(Figure 5g). The rise in the productivity at higher temperatures of ≥30–37 °C was also accompanied by the production of more than 80% antibodies as inclusion bodies (data not shown). In both regions of *E. coli*, the fermentation temperature had a great effect on human and rabbit scFvs and VHH productivity and impacted the solubility.

### 3.3. Purification of Soluble Antibody Fragment from E. coli Using Protein L Chromatography

The humscFv (Kz52), without non-canonical disulfide bonds, with a 6xHis tag was expressed in 500 mL bacterial culture under IPTG induction conditions to direct the expression in cytosolic and periplasmic compartments with a fermentation temperature of 25 °C. However, the overexpression resulted in the manufacture of soluble humscFv (Kz52) as well as the formation of inclusion bodies, as more than 50% of humscFv-kz52 was found in the insoluble pellet. The soluble scFv (Kz52) was isolated by lysing the cell pellet in Hepes buffer as described in method Section 2.5. After centrifugation of the cell-lysate, the cleared lysate was subjected to Protein L affinity chromatography by using the ÄKTA™ pure purification system (Figure 6). The purified fractions were evaluated by separating on SDS-gel electrophoresis. After staining the SDS gel with coomassie brilliant blue, prominent bands were observed at approximately 27 kDa (Figure 6c). The scFv from the periplasmic space showed the most intense band compared to cytosolic soluble scFv antibodies.

### 3.4. Solubilization and Purification of Refolded Inclusion Bodies by Direct Dilution Method

The recombinant antibodies (humscFv, RabscFv, and VHH), including classical and non-classical disulfide bonds, were expressed in *E. coli* at normal bacterial growth conditions. Mostly, in normal growth conditions, the major fraction of recombinant proteins was expressed in the form of non-classical IBs. The resultant IPTG induced bacterial cell pellets were solubilized by non-denaturing conditions as described in method Section 2.5. The insoluble fraction was used to isolate inclusion bodies as described in method Section 2.9. The resulting IBs were solubilized with moderate denaturing agents containing 6 M urea with 10 mM DTT, and then slowly dialyzed against buffers, as described in method Section 2.9.1, containing decreasing amounts of urea with the lowering of the DTT concentration. This process efficiently refolds the recombinant antibodies and any unfolded/insoluble material is removed by a centrifugation step. The solubilized rAbs were subsequently purified using a nickel chelating resin-affinity chromatography via the His-tag engineered on its C-terminus (Figure 7). The purified, refolded antibody fractions were separated on SDS-PAGE electrophoresis. After staining the SDS gel with coomassie brilliant blue, prominent bands were observed at approximately ~27 kDa for scFv and 15 kDa for VHH, respectively (Figure 7a–d). To monitor the possible structural modifications of the refolded protein, we performed a dot-blot analysis to detect the C-terminal c-Myc tag epitope. The antibodies refolded by the dialysis method showed weak immunoreaction as compared to partially soluble fractions (Figure 7a–d).

The purified proteins were dialyzed with PBS or Hepes buffer and ELISA assay against target antigens was performed. The experiment was performed by utilizing the same concentration of periplasmic soluble antibodies and refolded cytoplasmic protein. ELISA results also showed very weak immunoreactive signals as compared with soluble periplasmic antibody factions with similar antibody concentration (data was not shown). These ELISA and dot-blot results stated that the IB-solubilized refolded antibodies contain heterogeneous mixtures of both correctly folded and misfolded population.

### 3.5. Recovery of Biologically Active Abs from the Inclusion Bodies Using Matrix-Associated Protein Refolding

The active soluble antibodies were recovered from classical and nonclassical IBs using matrix-associated protein refolding after solubilizing the inclusion bodies in PBS, 500 mM NaCl, 6 M urea, 10 mM imidazole, 400 mM L-arginine, 3 mM GSH, 0.9 mM GSSG (ratio 3:1), pH 8.0. The cleared supernatant was loaded on to the nickel chelating resin to bind the antibodies via the His-tag engineered on its C-terminus, for the “on-column refolding” process as described in the methods Section 2.9.2. To correctly refold the soluble antibody that retains the biological activity, the antibody was immobilized onto a matrix (nickel resin) (Figure 8) and then slowly washed with wash buffers containing decreasing amounts of urea, along with L-arginine, a redox pair of oxidized with reduced glutathione and DTT.

This process efficiently refolds the highly conserved disulfide with proper canonical and non-canonical disulfide bonds. We also observed that in 1 L baffled shake flask culture, approximately ≥100 mg of soluble active antibody can be routinely purified. The refolded and purified antibody fractions were separated on SDS-PAGE electrophoresis. After staining the SDS-PAGE gel with coomassie brilliant blue, prominent bands were observed approximately at 27 kDa (Figure 8a,c) for rabbit and human scFvs and 15 kDa for VHH domain (Figure 8b). The purified protein fractions were subjected to dot-blot analysis to monitor the proper folding of the antibodies by probing the c-Myc tag epitope (Figure 8a–c). The purified protein was tested for its activity such as binding to recombinant protein by ELISA test. In addition, the purified Kz52 Ab was utilized for virus entry blocking assay using a dose-dependent antibody concentration as described in detail in our recently published group paper [31] as shown in Figure 8d.

## 4. Discussion

Recombinant production of high disulfide bond-rich proteins is challenging by the use of microbial hosts, as it results in the formation of inclusion bodies [31]. Morphologically, inclusion bodies are dense, amorphous protein deposits that can be found in both the cytoplasmic and periplasmic regions of bacteria [32]. The right refolding processes of inclusion body refolding processes are poised to play a major role in the production of recombinant proteins. Protein refolding is the process by which an unstructured protein alters its conformation to reach its native structure. Hence, the renaturation of these inclusion body proteins is a field of increasing interest in gaining large amounts of correctly folded proteins.

Improving biologically active renatured protein yields by minimizing aggregation and cost-effectiveness involves improvement in the process of inclusion body isolation, washing, solubilization of the aggregated protein, and actual refolding. While the efficiency of the washing and solubilization steps can be relatively high, refolding protein yields may be limited by the accumulation of inactive misfolded as well as aggregated protein [33]. Because the majority of cysteine-rich proteins including antibodies contains one or more disulfide bonds, refolding with concomitant disulfide-bond formation is challenging in a bacteria host [34]. Bacterial cells containing inclusion bodies are usually disrupted by high-pressure homogenization or a combination of mechanical, chemical, and enzymatic methods [35]. The resulting suspension is treated by either low-speed centrifugation or filtration to remove soluble proteins from the particulate containing the inclusion bodies. To date, no information was provided about the efficiency of IBs solubilization process or the range of inclusion body protein. The most commonly used solubilizing agents are denaturants, such as guanidinium chloride (GdmCl) and urea (6–8 M), and protein concentrations of 1–10 mg/mL during the solubilization step [36]. The solubilization process of IBs also requires a reducing agent such as dithiothreitol (DTT) or 2-mercaptoethanol to maintain cysteine residues in the reduced state in the presence of low molecular weight thiol reagents. These agents reduce non-native inter- and intra-molecular disulfide bonds possibly formed by air oxidation during cell disruption and will also keep the cysteines in their reduced state to prevent non-native intra- and inter-disulfide bond formation in highly concentrated protein solutions at an alkaline pH range of pH 8–9 [37]. After solubilization of IBs using a combination of reducing agents and high concentrations of denaturants, renaturation is then accomplished by the gradual removal of denaturants by different methods such as dilution, a buffer-exchange step such as dialysis, diafiltration, gel-filtration chromatography, or immobilization onto solid support [20,38]. However, conventional refolding methods, such as dialysis and dilution, are time-consuming and, often, recovered yields of active proteins are low, and a trial-and-error process is required to achieve success. The solubilization and matrix-associated refolding and purification process could present proper structure and function to the antibodies and is facilitated by binding through fusion partners, such as a His-tag [39] or the cellulose-binding domain [40], which retain their binding capabilities under the denaturing conditions required for loading the solubilized inclusion body protein onto the column.

Optimization of expression of soluble, active antibody fragments in the *E. coli* expression system was carried out by monitoring various parameters such as host strains, vector promoters, protein translocation tags, induction conditions, and purification strategies. The bacterium *E. coli* parental B strains such as BL2l derivatives and some derivatives of the K-12 lineage were used to check for the best outcome. The production of mostly soluble antibody fragments in the periplasmic space of *E. coli* is an established strategy in which the N-terminal bacterial pelB signal peptide targets the protein to the periplasmic secretory pathway [41]. The VH and VL antibody fragments have canonical disulfide bonds that are formed by an oxidative reaction in which the thiol (RSH) groups of cysteine pairs in VH and VL are joined by a covalent linkage forming canonical disulfide bonds. These structures contribute to the three-dimensional structure of an antibody and are often required for rendering the maximum biological activity to an antibody [42]. Conversely, llama, Camelid, Bovine, and rabbit antibodies, often human and mouse antibody proteins, have extra cysteine residues that are converted to non-canonical disulfide bonds [43]. Generally, during the expression of antibodies with disulfide bonds in the microbial system, final folded state proteins are translocated across the cytoplasmic membrane into the periplasmic, a redox environment with foldases (e.g., disulfide isomerases (Dsb systems)) that catalyze the formation of disulfide bonds [44]. Importantly, our study shows that metal affinity associated refolding of Abs including additives such as L-arginine with the redox pair of oxidized and reduced glutathione, resultant antibodies were retained biological activity with a high soluble Abs yield. Oxidation can also be achieved by adding a mixture of oxidized and reduced thiol reagents such as glutathione, cysteine, and cystamine [23,45]. Renaturation with mixed disulfide bond formation using oxidized glutathione also helps in the high recovery of disulfide-containing protein. This involves the formation of disulfide bonds between glutathione and the denatured protein followed by renaturation in the presence of a catalytic amount of reduced glutathione [22,37]. The use of mixed disulfides increases the solubility of the protein during refolding and thus helps in lowering the extent of incorrect disulfide bond formation and amount of aggregation protein.

In this study, we employed different bacterial host strains to produce high cysteine-rich antibody fragments mainly utilizing engineered *E. coli* Rosetta-gami B(DE)3 [46], and Shuffle^®^T7 Express [47]. These *E.coli* strains were developed to correctly fold disulfide-bonded proteins in the cytoplasmic space and were successfully used for producing biologically active antibody fragments and IgG [15] Notably, we observed enhancement in the production of disulfide bond-containing antibody proteins in the periplasmic and cytosol space using the Shuffle^®^T7 Express bacterial strain compared to other strains. These differences could potentially be explained by the engineered ΔtrxB and Δgor adding a signal sequence less than DsbC that is constitutively expressed in the cytoplasmic space and enhances the production of disulfide-rich proteins [48,49]. Notably, we also noted that the overexpression of periplasmic targeted Abs in *E. coli* may lead to toxicity and low yields of the active, soluble antibodies as previously described [50]. The color of harvested pellets changed depending on the toxicity level prompting partial cell-population death during the culture conditions (Figure 4c,d) The gene fusion technology has also widely been used to improve the soluble protein production, and our observation indicated that the yield of the soluble N-terminus thioredoxin (TrxA) tagged Ab production was augmented; however, the biological activity of the antibodies was compromised. Therefore, the removal of the tag seemed essential to recover the biological activity of the antibodies. Indeed, it is a harsh, time-consuming, and expansive process, which would greatly hamper overall recovery and yield by an acceleration of re-aggregation as previously described [51]. Frequently, we found that a higher percentage of desirable Ab fragments were formed in the insoluble, inclusion bodies despite using different *E. coli* strains for targeting the protein expression in periplasmic or cytoplasmic compartments. A high number of disulfide bonds is very challenging as often, almost exclusively, it is known to aggregate in inclusion bodies [52]. Hence, we detected a more aggregated form of rabbit scFv protein with non-canonical disulfide bonds compared with human scFv (Kz52) without non-canonical disulfide bonds. To understand the IB derived solubilized protein folding, c-Myc binding epitope-based dot-blot immunoreaction and ELISA based antibody paratope binding analysis were performed. The results (Figure 8) indicated that solid support refolding purification allows slow–steady refolding that creates a favorable environment to correctly fold the antibody irrespective of the presence of non-canonical disulfide bonds and showed biological activity. On the other hand, the immunoblot data suggested that the dialysis method employed for refolding the protein yielded a heterogenous form (correct and misfolded protein). The diminished dot-blot signal may be due to the structural modification resulting in the masking of the C-terminus which contains the c-Myc tag epitope.

In conclusion, there is no unique microbial host system, yeast, or mammalian cell-system that can consistently provide a high yield of expressed antibody fragments for a wide range of single-chain antibodies. It is the product sequence and end application that regularly dictates the optimum choice of expression host. Notwithstanding host systems, adequate preceding design and engineering of the molecular construct supported by a good biophysical understanding of the single-chain fragment molecules are crucial pre-requisites for improved product stability and downstream recovery, which will favorably impact final product yield and functionality. Overall, we believe that the strategy presented in this study could also be adapted to produce other valuable mammalian disulfide bond containing proteins in the *E. coli* expression system. It is expected that a novel solubilization process which reduces the propensity of protein aggregation followed by improved refolding will help in the high recovery of recombinant protein from inclusion bodies. Domain antibodies with multiple disulfide bonds need a more elaborate refolding process in the presence of optimal concentrations of both oxidizing and reducing agents for the formation of proper disulfide bonds.

## Figures and Tables

**Figure 1 antibodies-09-00039-f001:**
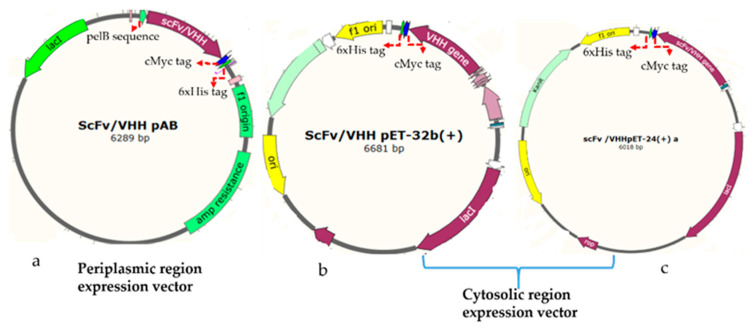
Schematic presentation of the antibody gene cloning vectors for the expression in the periplasmic and cytosolic compartments in the microbial system. (**a**) Design of cloning vector for the expression of the antibody fragment in the periplasmic space, (**b**) design of cloning vector for the expression of the antibody fragment in the cytosolic space with a thioredoxin tag as a chaperone protein in an oxidative cytosolic environment. (**c**) Design of cloning vector for the cytosolic space expression of the antibody fragment; the 6xHis tag and cMyc tag serve as an affinity purification tag and a signal detection tag, respectively. The plasmid DNA maps were generated using SnapGene software.

**Figure 2 antibodies-09-00039-f002:**
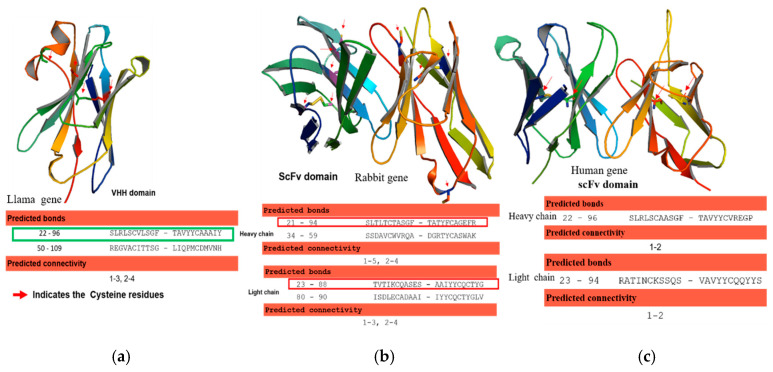
The schematic representation of homology structures of the antibody domain and possible canonical and noncanonical disulfide bond formation. (**a**) Homology model of camelid single-domain (VHH) llama antibody domain showing the predicted disulfide bond; (**b**) single-chain fragment variable (scFv) domain of rabbit antibody (RabscFvAb) homology model showing the number of possible disulfide bonds; (**c**) single-chain fragment variable (scFv) domain of the human antibody (humscFvAb) domain homology model showing the disulfide bonds. Red color arrows indicate the cysteine residues that are involved in the possible disulfide bonds as shown in the diagrams.

**Figure 3 antibodies-09-00039-f003:**
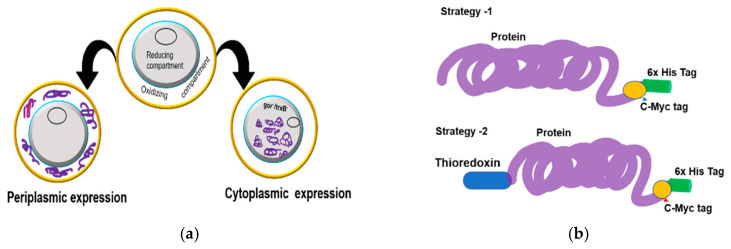
Schematic presentation of the antibody expression in the microbial system. (**a**) Depiction of highly conserved disulfide-bonded antibody fragment in the periplasmic and cytosolic compartments; (**b**) design of the antibody fragment with a thioredoxin tag as a chaperone protein for expression in an oxidative cytosolic environment. The 6xHis tag and c-Myc tag serve as an affinity purification tag and a signal detection tag, respectively.

**Figure 4 antibodies-09-00039-f004:**
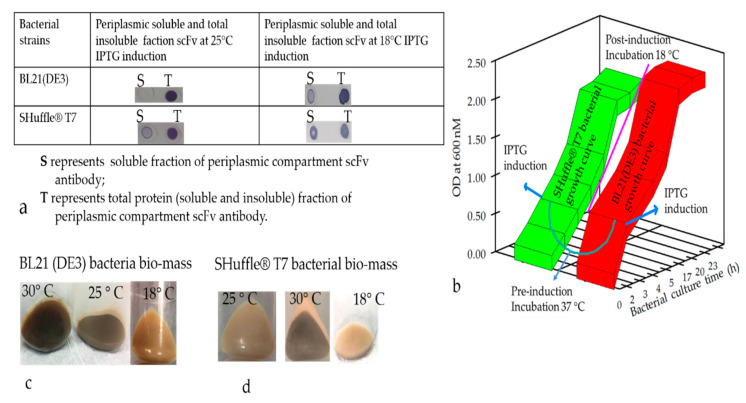
Expression pattern of the antibody fragments at different temperature conditions. (**a**) Dot-blot analysis of soluble and insoluble antibody proteins from induced bacterial cell pellets from 2 bacterial strains. The soluble antibody fraction and total antibody proteins were isolated and spotted onto the nitrocellulose membrane. The presence of antibodies in the spots was examined by using horseradish peroxidase (HRP) tagged anti-myc monoclonal antibody, followed by a short incubation with Tetramethylbenzidine (TMB) substrate. (**b**) Pictorial depiction showing the growth curve of two bacterial strains at two different temperatures. (**c**) Image showing the color of the bacterial cell pellets expressing antibody fragments in periplasmic and cytosolic compartments. Three batches of bacterial cultures were induced at the same density (O.D of 0.8) but at 3 different temperatures as shown in the figure and pelleted after 14 h.

**Figure 5 antibodies-09-00039-f005:**
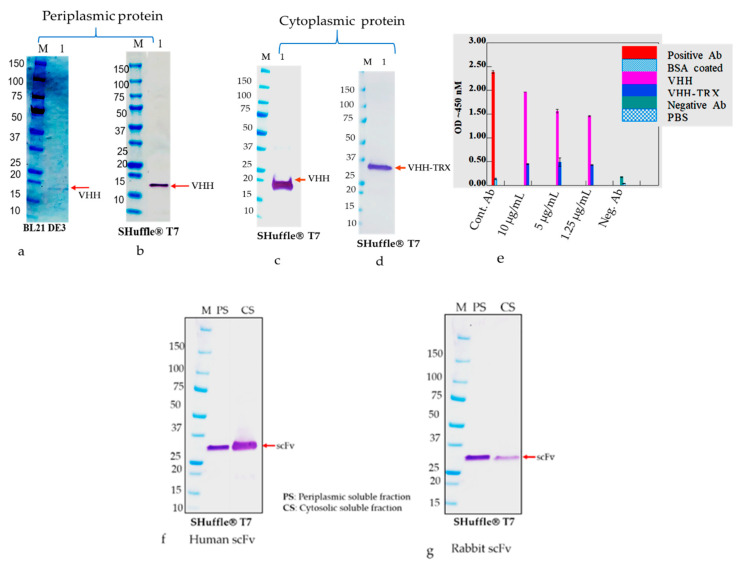
Monitoring expression and binding activity of antibody fragments using Western blot analysis and an ELISA test from periplasmic and cytosolic compartments of *E. coli.* (**a**) Periplasmic expression of antibody fragment (VHH) in BL21(DE)3; (**b**) periplasmic expression antibody fragment (VHH) in SHuffle^®^ T7; (**c**) cytosolic expression of soluble VHH antibody without TrxA; (**d**) cytosolic protein with thioredoxin (TrxA) as a disulfide reduction, notice that the molecular weight is approximately 26 kDa. (**e**) Figure 5e represents the binding of the antibody with/without the TrxA tag on VHH protein from cytosolic expression in *E. coli* as described in method Section 2.9. The ELISA standard deviation values of the ELISA signal values were in triplicates. (**f**,**g**) Representation of periplasmic and cytosolic expression of human and rabbit antibody fragments (scFvs) using Shuffle^®^ T7. The soluble antibody fractions were isolated, run on SDS-PAGE, and transferred to nitrocellulose membrane. The Western blots of antibody proteins were performed by using anti-myc HRP tagged monoclonal antibody and the specific immunoreactive bands were visualized using TMB substrate.

**Figure 6 antibodies-09-00039-f006:**
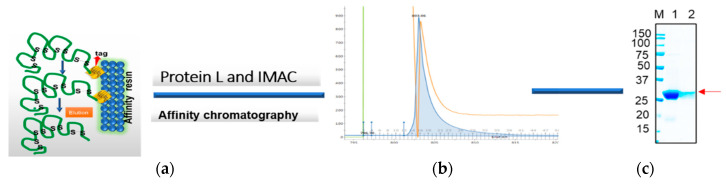
The affinity chromatography purification of the scFv fragment using protein L resin. The soluble antibody fragment was isolated from isopropyl β-d-1-thiogalactopyranoside (IPTG) induced bacterial cell pellet and subjected to affinity chromatography using an immobilized metal affinity matrix (IMAC) or a protein L column (binds to the kappa light chain of the antibody); (**a**) represents a schematic illustration of the chromatography. (**b**) A representative chromatogram peak image. (**c**) Eluted fractions were analyzed by 4–20% gradient SDS-PAGE under reducing conditions. Lane MW: molecular weight; lane 1: purified human scFv from periplasmic proteins; and lane 2: purified human scFv from cytosolic proteins.

**Figure 7 antibodies-09-00039-f007:**
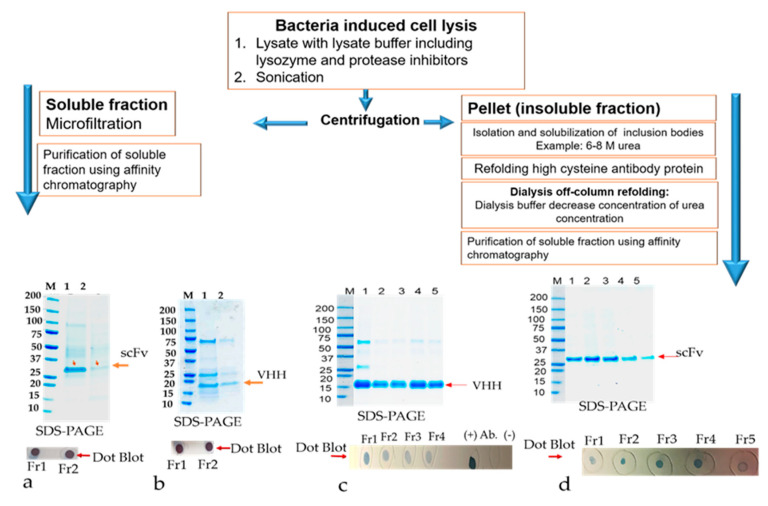
The affinity chromatography purification of rabbit scFv and VHH fragments using immobilized metal affinity chromatography (IMAC) after Abs refolding by the dilution method as described earlier. The soluble and IBs solubilization antibody purified fractions were loaded to the IMAC column, and bound proteins were eluted and analyzed by 4%–20% gradient SDS-PAGE under reducing conditions. (**a**,**d**) SDS-PAGE gel profile of the purified refolded fragment of RabscFv; (**b**, **c**) SDS-PAGE gel profile of purified VHH domain. The eluted fractions were analyzed by dot-blot analysis on the nitrocellulose membrane. The dot-blots of antibody proteins were performed using an anti-cmyc-HRP tagged monoclonal antibody, and the specific immunoreactive bands were visualized using TMB substrate. (**a**,**b**) Figure 7a,b represent soluble fraction Abs purified fractions, lane M indicates molecular weight, lanes 1–2 indicate purified fractions of interest of protein; RabscFv and VHH, respectively. Figure 7c,d represent IBs solubilized Abs purified fractions, lane M indicates molecular weight, lanes 1–5 indicate purified fractions of interest of protein; VHH and RabscFv, respectively. Fr. indicates the elution fraction. Ab (+) indicates the control antibody. Ab (−) indicates PBS buffer.

**Figure 8 antibodies-09-00039-f008:**
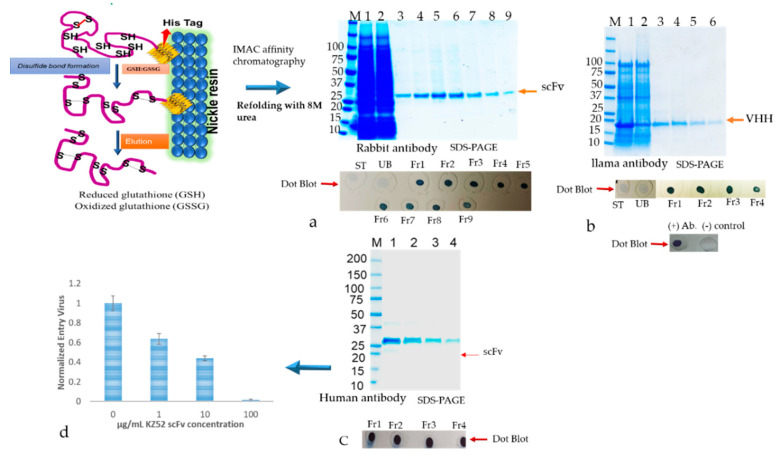
The affinity chromatography purification of scFv and VHH fragments using immobilized metal affinity chromatography (IMAC) after refolding of antibody fragments as described in method Section 2.9.2. (**a**) The refolded antibody fragment protein was subjected to IMAC as described in detail in the method section, and eluted fractions were analyzed using 4–20% gradient SDS-PAGE under reducing conditions. (**a**–**c**) Coomassie-stained SDS-PGE gel profile of purified refolded antibody fragments. Dot-blots are also shown at the bottom of the gel image. (**d**) Biological activity of the post-purified antibody, showing blockage of Ebola virus entry. The mean with standard deviation values of the ELISA were in triplicates. Lanes M indicates molecular weight, lanes 1–5 indicate purification fractions of Abs, respectively. ST represents starting material and UB represents unbound fractions. Fr indicates fraction. Ab (+) indicates control antibody. Ab (−) indicates PBS buffer.

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
