# Peer review of "Optimization of Methods for the Production and Refolding of Biologically Active Disulfide Bond-Rich Antibody Fragments in Microbial Hosts"

_2073-4468, 2020, doi:10.3390/antib9030039_

Round 1

Reviewer 1 Report

Manuscript number: antibodies-857130

Optimization for the Production of Heterologous Disulfide Bonded Antibody Fragments in the Periplasmic and Cytoplasmic Compartments of E. coli

Bhupal Ban, Jagathpala Shetty

Authors optimized the production protocol for human serum albumin specific single domain antibody(VHH), monoclonal rabbit antibody as an scFv format with anti-Ebola specific activity. They used moderate-solubilization and in vitro matrix associated refolding strategies with redox pairing and found better structured antibodies production. It yielded functionally more active antibody fragments than the one achieved by in vitro refolding method. It is very interesting, but needs major revision for showing that they demonstrate and logically explain the evidences and results that support this opinion.

 1. The optimization process should be shown with comparison data stepwise. The final best condition should be shown with the other results. And please compensate discussions for the reason and mechanism of refoling process.

2. Table 1 conditions are not novel and it needs correction and matched explanation for Fig 3. Fig 3 needs new formatting to show clearer results with matching conditions. Fig 3a is especially difficult to read.

3. Is ELISA results of Fig 4 triplicates? then show the standard deviation bar. Moreover the positive control (using comercally available antibody standard) data is necessary.

4. Fig 6 & 7 needs also positive control data to show the clear evidence of activity of refolded antibodies if it is possible.

5. Please explain more about “more elaborate refolding process” with theoritical explanation and data or references in discussion.  

6. Change the title so that you can express your research more specifically.

Author Response

The optimization process should be shown with comparison data stepwise. The final best condition should be shown with the other results. And please compensate discussions for the reason and mechanism of refolding process

In this manuscript, the main aim is to perform a comparative study on numbers of disulfide bonds present in the antibodies on the expression in the microbial system, to determine if we can optimize a method towards expression and purification of antibodies.   Here, we demonstrated that the relative number of cysteine residues involved in both canonical and non-canonical disulfide bonds in an antibody sequence can have a direct influence on the production of soluble antibody fragments.  The higher number of cysteines have a negative effect on the production of soluble antibodies and the effect is even more undesirable if the disulfide bonds formed are non-canonical possibly due to interexchange of disulfide bonding during the expression in the bacterial system (Fig. 5g) This tendency was observed mostly in the antibodies expressed in the cytoplasmic compartment.  Our results also demonstrated that the refolding of desirable antibodies from the bacterially expressed insoluble antibodies required an alkaline pH range of 8-9 to prevent non-native intra-and inter-disulfide bond formation in highly concentrated condition. Importantly, we introduced a simple method to understand the protein refolding from IBs-derived, solubilized, antibody fragments by using dot-blot and ELISA based immunoreaction method. These results indicated that solid support refolding and purification of an antibody (with or without non-canonical disulfide bonds) allow slow, steady refolding that create a favorable environment to correctly fold the antibody with desired biological activity. The optimization of methods for expression and purification are not constantly linear, importantly, it depends on properties of amino acids residues, engineering of the molecular construct supported by the good biophysical understanding of antibodies, which is a crucial pre-requisite for improved product stability and downstream recovery, and favorably impact final product yield and functionality.      

These results indicated that solid support refolding purification allows slow-steady refolding that creates a favorable environment to correctly folded antibody with/without non-canonical disulfide bonds with biological activity. The method optimization for expression and purification are not constantly liner and steady, importantly, it depends on properties of amino acids residues, engineering of the molecular construction supported by a good biophysical understanding of antibodies, is a crucial pre-requisite for improved product stability and downstream recovery, which will favorably impact final product yield and functionality. We believe that this manuscript educates the reader in the domain antibody research filed.       

  1. Table 1 conditions are not novel and it needs correction and matched explanation for Fig 3. Fig 3 needs new formatting to show clearer results with matching conditions. Fig 3a is especially difficult to read.

We deleted table1. The figure 3 (the current Fig.4) reformatting you can see in this manuscript. 

  1. Is ELISA results of Fig 4 triplicates? then show the standard deviation bar. Moreover, the positive control (using commercially available antibody standard) data is necessary.

Yes, the ELISA result was carried out triplicates, also we included positive control. Current ELISA result representant in Fig.5e.

  1. Fig 6 & 7 needs also positive control data to show clear evidence of the activity of refolded antibodies if it is possible.

We modified the previous Fig.6 and Fig.7 (current version, Fig. 7 and Fig.8.). We added a simple and quick testing method to understand the IBs derived, solubilized protein refolding, c-Myc binding epitope-based dot-blot, prior to ELISA test. We noted that solid support refolding purification allows slow-steady refolding that creates a favorable environment to make a correctly folded antibody (with/without non-canonical disulfide bonds) with biological activity. we are preparing another manuscript using specific binding data, therefore, in this manuscript, we are unable to include antigen-specific data of the rabbit antibody.    

3. Change the title so that you can express your research more specifically.

Optimization of Methods for the Production and Refolding of Biologically Active Disulfide Bond-Rich Antibody Fragments in Microbial Hosts”.

Reviewer 2 Report

The manuscript is focused on the optimization of the protocol for the production of antibody fragments in E.coli.

E.coli is one of the first choice systems for the production of recombinant proteins for the high yield obtained and easiness to use. One of the principal drawbacks of this system, besides the impossibility to have further modifications of the protein, is the difficulty, sometimes encounter, to obtain soluble and active proteins. For this reason, usually, the expression of the antibodies is conducted in superior systems like Yeast and Mammalian cells where however the protein production is lower. For this reason, it is important to investigate more protocols and strategies to obtain soluble and active proteins, especially recombinant antibodies, in the E.coli system.

However, considering the present study, I have some doubts regarding the novelty of the strategies used and the results obtained. The periplasmic expression of recombinant proteins is very well known for many years as well as the different conditions of temperature and inducer concentration during the expression, and the matrix associated protein refolding using reducing agents to promote the formation of disulfide bonds. In my opinion, the study summarizes the possible strategies to be used for the production of active antibody fragments in E.coli. If it were not for the experiments conducted the manuscript could be a sort of review of the "Production of Heterologous Disulfide Bonded Antibody Fragments in the Periplasmic and Cytoplasmic Compartments of E.coli" since the strategies used have been described also by other authors.

The study is well written and well organized, some parts can be improved to better understand all the experiments conducted:

  • Paragraph 2.3: the authors have to better describe the origin of the rabbit monoclonal antibody, consistently with the description of the other antibodies.
  • Paragraph 2.3_ Lane 124: please explain the meaning of PDB: 3CSY
  • Paragraph 2.4: the authors have to better describe the feature of the plasmids used to express the antibodies. In particular, the pelB leader permits to lead the protein in the periplasmic space but from the manuscript is not clear. If I understand correctly when the authors talk about periplasmic expression they are referring to all the antibodies expressed in the plasmid with pelB leader otherwise when they refer to cytosolic means the expression in one of the other two vectors. I think that, if it so, it should be explained more clearly. In this way also the table 1 will result more clearly.
  • Paragraph 3.1: at the beginning of the paragraph, where it is described the isolation of antibody specific genes, the human-derived antibody gene is not mentioned.
  • Paragraph 3.2- lane 137: is "The scFv" referred to both the scFv antibodies fragments expressed?
  • Paragraph 3.2: sometimes when the authors mention the scFv is not very clear to which scFv they are referring to. If the behavior of the two scFv expressed is similar please indicate it.
  • 4. Please indicate from which antibody the scFv fragment has been originated.

Minor considerations:

  • Line 118: HAS meaning has to be included here and not at lane 121
  • Line 128: Please change "is" with "was" in the sentence
  • Line 279: why mouse genes are mentioned since no mouse antibodies are included in the study?

Author Response

Authors answer

To address reviewer comments and suggestions about this manuscript, commonly, antigen-binding domain sequences do not necessarily require further posttranslational modification to retain its binding properties, therefore, it is very useful hosts to get desirable reagents using the microbial system. Also, yeast and mammalian cells are growing slow, highly sensitive, and time-consuming such as longer doubling time, most importantly, yeast and mammalian system required specific instruments and more costly to maintain cell growth culture media as well as cell lysate reagents. Conversely, antibody molecules are highly diversified amino acids including a number of cysteine residues, eventually, biophysical properties of each antibody molecule are a crucial pre-requisite for improved product stability, downstream recovery, and end application often dictate the optimum choice of expression host. Here in this manuscript, we are focusing mainly on design, host selection, and final recovery of quality not quantity antibody with desirable activity in both compartments; periplasmic and cytosolic regions. The periplasmic expression system is well-known; however, 50- 60 % desirable proteins are still located in the cytosolic region as inclusion bodies, therefore, refolding of inclusion antibodies are urgent need to recover in cheaper laboratory method. We have demonstrated that there is no direct method that applies to all antibody molecules production and purification in bacterial as well as other expression hosts. We also noticed that all recovery proteins are not quality reagents to recover binding properties. Of course! It is not a completely new method, nonetheless, we validated literature-based methods using different antibodies and adapted the dot blot method to monitor folding antibody-protein properties prior to biological activity. This manuscript provides the reader a clear picture of design, production, and purification methodology, importantly, give a good reason for developing a method added a significant value such as save time and lowering cost for the production of heterologous sequence-based antibodies reagents.    

 The study is well written and well organized, some parts can be improved to better understand all the experiments conducted:

  • Paragraph 2.3: the authors have to better describe the origin of the rabbit monoclonal antibody, consistently with the description of the other antibodies.

Authors elaborate on all antibody generation methods including related references are listed, but we are unable to include the sequence in this manuscript.  

  • Paragraph 2.3_ Lane 124: please explain the meaning of PDB: 3CSY

The human-derived anti-Ebola virus (Kz52) heavy and light chain variable genes were obtained from the Ebola virus glycoprotein in complex with a neutralizing antibody crystal structure ID: 3CSY at Protein Data Bank (PDB).

  • Paragraph 2.4: the authors have to better describe the feature of the plasmids used to express the antibodies. In particular, the pelB leader permits to lead the protein in the periplasmic space but from the manuscript is not clear. If I understand correctly when the authors talk about periplasmic expression, they are referring to all the antibodies expressed in the plasmid with pelB leader otherwise when they refer to cytosolic means the expression in one of the other two vectors. I think that, if it so, it should be explained more clearly.

Authors insert in detail cloning and selection of cloning vectors under section 2.4. Designing and cloning of Abs genes with and without fusion tag lanes 144 to 169 including inserted cloning plasmids map as Fig.1.

  • Paragraph 3.2- lane 137: is "The scFv" referred to both the scFv antibodies fragments expressed?

We described all scFvs fragment name HumscFv, RabscFv, respectively.

  • Paragraph 3.2: sometimes when the authors mention the scFv is not very clear to which scFv they are referring to. If the behavior of the two scFv expressed is similar please indicate it.

We evaluated soluble antibodies production using two scFv constructs one containing canonical disulfide bonds (HumscFv), and the other containing non-canonical disulfide bonds (RabscFv), (Fig.5f and 5g).  we noticed that the relative increase in the number of non-canonical disulfide bonds had a concomitant negative effect on the production of soluble proteins in both compartments, mostly in the cytoplasmic region.

Round 2

Reviewer 1 Report

Authors revised the manuscript according to the comments.